# Third-Stage Dispersal Juveniles of *Bursaphelenchus xylophilus* Can Resist Low-Temperature Stress by Entering Cryptobiosis

**DOI:** 10.3390/biology10080785

**Published:** 2021-08-17

**Authors:** Long Pan, Rong Cui, Yongxia Li, Wei Zhang, Jianwei Bai, Juewen Li, Xingyao Zhang

**Affiliations:** 1Research Institute of Forestry New Technology, Chinese Academy of Forestry, Beijing 100091, China; Longpan@caf.ac.cn (L.P.); cuibaobei93@126.com (R.C.); zhangwei1@caf.ac.cn (W.Z.); xyzhang@caf.ac.cn (X.Z.); 2Research Centre of Sub-Frigid Zone Forestry, Chinese Academy of Forestry, Harbin 150080, China; 3Co-Innovation Center for Sustainable Forestry in Southern China, Nanjing Forestry University, Nanjing 210037, China; 4Chongqing Forestry Investment Development Company Limited, Chongqing 401120, China; zilin0828@foxmail.com; 5Graduate Department, Chinese Academy of Forestry, Beijing 100091, China; lijuewen0315@126.com

**Keywords:** *B. xylophilus*, cryptobiosis, osmotic regulation, low-temperature stress

## Abstract

**Simple Summary:**

Pine wilt disease caused by the nematode *Bursaphelenchus xylophilus* causes significant harm to China’s forests, but there are currently no effective prevention and control measures. Additionally, this devastating disease is currently spreading northward. We determined that third-stage dispersal juveniles of *B. xylophilus* can resist low-temperature stress by cryptobiosis, allowing these nematodes to tolerate a greater range of temperatures. These results facilitate the prediction of potential areas at risk for *B. xylophilus* in the mid-temperature and cold temperature zones of China.

**Abstract:**

Nematodes can enter cryptobiosis by dehydration as an adaptation to low-temperature environments and recover from cryptobiosis by rehydration after environmental improvement. In this work, the survival of *Bursaphelenchus*
*xylophilus* third-stage dispersal juveniles was studied in response to low-temperature treatment. The average survival rates were 1.7% after −80 °C treatment for 30 d and 82.2% after −20 °C treatment for 30 d. The changes of water content and inorganic salt ions that occur in pine trees during winter gradually alter the osmotic pressure in the liquid environment to dehydrate *B. xylophilus* juveniles, resulting in improved survival after low-temperature treatment. The survival rate at −20 °C improved to 92.1% when the juveniles entered cryptobiosis by osmotic regulation. The results of this study demonstrate that *B. xylophilus* third-stage dispersal juveniles can resist low-temperature stress through cryptobiosis, providing the theoretical basis for the identification of areas potentially vulnerable to *B. xylophilus* in the mid-temperature and cold temperature zones of China.

## 1. Introduction

Pine wilt disease caused by *Bursaphelenchus xylophilus* is one of China’s most damaging forest diseases. *Bursaphelenchus xylophilus* has recently expanded from southern China to northern areas and is now threatening new hosts, such as *P**inus tabuliformis, P**. koraiensis,* and *P**. sylvestris* [1]. In 2017, a large area of *P. koraiensis* in Dandong City of Liaoning province suffered significant damage due to pine wilt disease [2]. This was the first identification of this disease in this monsoon-controlled mid-temperature zone with an annual average temperature of 2–8 °C, indicating that *B. xylophilus* has expanded its habitat. In this region, the air temperature gradually starts to decrease in the autumn, with a minimum temperature in winter that is below −20 °C [3]. The internal temperature of pine trees declines with the air temperature, causing the metabolic rate to slow. Additionally, the water content gradually declines and the concentration of potassium ions increases [4]. Overall, these changes can reduce contact between *B. xylophilus* and water, limiting freezing and osmotic stress to *B. xylophilus* [5].

In response to osmotic stress, numerous nematodes can enter cryptobiosis, a physiological state in which metabolic activity is reduced. There are recent reports describing cryptobiosis of nematodes from over 30,000 years ago [6,7,8]. Through cryptobiosis, nematodes can survive extreme environments, such as low temperature, and extend their lifecycle. These nematodes enter cryptobiosis gradually through dehydration [9]. After cryptobiosis, individuals may suffer from declined metabolism and suspended life cycle, but can recover through rehydration under suitable conditions [10,11]. Osmotic regulation controls the exchange of in vivo and in vitro substances during the dehydration and rehydration stages before and after cryptobiosis to achieve osmotic balance and adjust to crucial changes in water content and inorganic salt ions in the external environment. The ability to resist low-temperature stress is related to the in vitro environment and can be influenced by in vivo cold-resistant substances [12]. Previous studies report significantly increased in vivo content of trehalose and glycerin during the dehydration of nematodes [13].

The life history of *B. xylophilus* can be divided into the reproductive cycle and the diffusion cycle [14]. Under a suitable environment, *B. xylophilus* enters the reproductive cycle (EGG, J1, J2, J3, J4 and Adult) and the population increases continuously, resulting in the morbidity and death of pine trees. Under adverse environmental conditions, *B. xylophilus* enters the diffusive cycle, i.e., the diapause cycle, which consists of two stages: third-stage dispersal juveniles (JIII) and fourth-stage dispersal juveniles (JIV) [14]. When the pupae emerge into adult insects, the JIII nematodes are induced to form the JIV, which enters the body of *M**onochamus sinensis* [15]. When the emerged adult longhorns feed on healthy pines, the fourth instar transfers to healthy pines, transformed and developed into the reproductive type of adults, and enters the reproductive cycle to multiply and harm pine trees in large numbers [16].

Propagative *B. xylophilus* can resist −5 °C minimum temperature stress, but *B. xylophilus* can transform from a propagative type into third-stage dispersal juveniles under adverse environmental stress [17,18]. However, previous studies did not investigate the response of *B. xylophilus* to low-temperature stress [19]. This study demonstrates the adaptation of third-stage juveniles of *B. xylophilus* to a low temperature environment, allowing prediction of potential distribution areas.

## 2. Materials and Methods

### 2.1. Isolation of B. xylophilus from Different Hosts

Specimens of *B**. xylophilus* from different hosts were collected during winter in Cangshi Village of Fushun City (Liaoning province), when the daily minimum temperature in winter is approximately −10 °C. Samples of four different hosts, withered and dead *P. koraiensis*, *P. tabuliformis*, *P. sylvestris*, and *Larix olgensis* were cut from areas affected by *B. xylophilus*. Round logs with 5 cm thickness were collected from the top, middle, and bottom of the withered trees, and 36 samples were taken to obtain *B. xylophilus*. Each sample was 2–5 kg, and 500 g was taken for each experiment to isolate *B. xylophilus*. The quantity and density of *B. xylophilus* third-stage dispersal juveniles were calculated after removal using the Baermann funnel method [20].

After removal, we used a 10 μL pipettor to transfer nematodes to a slide. The cultured nematodes were placed over the flame of an alcohol lamp and gently heat killed. The nematodes were observed under an inverted microscope and identified according to morphological and taxonomic characteristics. The third-stage dispersal juveniles exhibit obvious spillage at the head tip and slender oral needles. The middle esophageal bulb is ovoid, accounting for about two thirds of the body width. The body is filled with a large number of fat granules, with a finger-shaped tail. Nematode DNA was extracted using a DNA preparation kit (AP-MN-MS-GDNA-50; Axygen, San Francisco, CA, USA) according to the kit instructions. To confirm identification, PCR amplification and sequencing were performed using specific primers for *B. xylophilus* ITS (upstream primer F1: 5 ‘-CGT AAC AAG GTA GCT GTA-3′ and downstream primer V2: 5 ‘-TTT CAC TCG CCG TTA CTA AGG-3′).

### 2.2. Survival of Bursaphelenchus Xylophilus Third-Stage Dispersal Juveniles in Pine Trees under Low-Temperature Stress

*Bursaphelenchus**xylophilus* was extracted from one round log of affected *P. tabuliformis* using the Baermann funnel method [4]. The initial density in this log was determined. The log was then divided into five equally sized pieces that were separately wrapped with paper and stored in freezers at different temperatures (−20 °C, −30 °C, −40 °C, −70 °C, −80 °C) for low-temperature treatment. Approximately 30 d, 60 d, and 90 d later, the density of *B. xylophilus* specimens in these samples was determined by counting the numbers of individuals in 500 g samples to calculate the survival rate. This test of survival after low temperature treatment was repeated five times.

### 2.3. Assessment of the Ability to Resist Low-Temperature Stress by Osmotic Regulation after Cryptobiosis

To investigate effects on osmotic regulation, we prepared solutions of 15% KCl, 8% KCl (Aladdin, P112133-500 g), 8% glycerin (Aladdin, G116202-5 mL), 8% trehalose (Aladdin, T100010-25 g), and 8% sucrose (Aladdin, S112226-500 g). Each solution was separately tested by application of 500 μL with a pipette to a concave glass side, and then 300 *B. xylophilus* juveniles were transferred from distilled water to this liquid using a 10 μL pipette. The liquid’s osmotic pressure gradually increased with water evaporation, causing gradual dehydration of the *B. xylophilus* over approximately 5 h. The *B. xylophilus* juveniles were then transferred to a −20 °C constant temperature environment. After 24 h, the samples were removed, water was applied to the glass side, and then the samples were sealed in a Petri dish (diameter of 12 cm). After 24 h, the survival rate of *B. xylophilus* juveniles was determined by counting under the microscope (Olympus, CKX53, ×4 ×10 ×40). This test was repeated five times.

### 2.4. Cryptobiosis by Dehydration of B. xylophilus Juveniles

To observe the dehydration and rehydration process of *B. xylophilus* during cryptogenesis, the same solutions described in 2.3 were used and dehydration was performed. The dehydration process was observed using a microscope, and photos were taken. For rehydration, 500 μL distilled water was dripped on the concave glass side at room temperature, and then the *B. xylophilus* rehydration process was observed using a microscope.

### 2.5. Statistical Analysis

For analysis and visualization of the results of all experiments, we used Microsoft Office Excel (2007). The standard deviation and significant differences of the data were calculated by SPSS 18.0 software (SPSS, Inc., Chicago, IL, USA). The *t*-test was used to compare the two groups of data, and differences between the two groups (*p* < 0.05) were represented by different letters.

## 3. Results

### 3.1. Population Characteristics of B. xylophilus in Different Pine Trees in Fushun during Winter

*Bursaphelenchus xylophilus* was isolated from dead logs of *P. tabuliformis, P. koraiensis,* and *P. sylvestris,* with an average density of 42, 10, and 15 nematodes/g of wood, respectively, and the PWN was not detected in *L. olgensis* logs. The maximum and minimum densities of *B. xylophilus* in logs were 731 nematodes/g of wood and 52 nematodes/g of wood, respectively. Of the *B. xylophilus* isolated from the logs of the different hosts, 95% were the third-stage dispersal juveniles (JIII).

### 3.2. Survival Rates of B. xylophilus in P. tabuliformis after Low-Temperature Stress

The survival rates of *B. xylophilus* in *P. tabuliformis* were measured after low-temperature stress (Figure 1). The *B. xylophilus* survival rate after treatment under −20 °C for 30 d was 82.2%. Survival rate gradually declined with a decrease in treatment temperature and an increase in treatment time. Some *B. xylophilus* survived under −80 °C, with a survival rate after 30 d of 1.7%. According to the experimental results, the *B. xylophilus* third-stage dispersal juveniles can survive in a host at −20 °C for a long time, with fewer surviving at −80 °C.

### 3.3. Survival Rate of Diffused the Third Stage Juveniles B. xylophilus under Low Temperature after Osmotic Regulation

After soaking the *B. xylophilus* third-stage dispersal juveniles in different solutions of osmotic substances, the dehydration test was again performed. There were differences in the survival of *B. xylophilus* juveniles dehydrated by water evaporation in solutions with initial concentrations of 15% KCl, 8% KCl, and 8% glycerol (Figure 2). The survival improved significantly after dehydration, with the highest survival rate in 8% KCl solution at 92.1%. These results indicate that the survival of *B. xylophilus* under low temperature was significantly improved after dehydration with different osmotic substances.

### 3.4. Cryptobiosis by Dehydration for B. xylophilus Third-Stage Dispersal Juveniles

To observe the dehydration and rehydration process of *B. xylophilus* during cryptobiosis, *B. xylophilus* third-stage dispersal juveniles were first soaked in a solution of 8% KCl. With the removal of water, the nematodes curled significantly (Figure 3B,C) and moved freely. Some KCl crystals could be seen when the water was evaporated completely, and at this point the nematodes stopped moving and clustered together (Figure 3D). Shrunken bodies, sunken body surface, and accumulated fat granules (Figure 3E,F) were observed. These dehydrated samples were subjected to low-temperature treatment, distilled water was added for rehydration, and recovery was observed (Figure 3A).

## 4. Discussion and Conclusions

Cryptobiosis, dormancy in a dehydration state, provides resistance to extreme environments and can extend the lifecycle [21]. This is a common strategy for nematodes. Some nematodes can decrease their in vivo water content to 3% (from a normal 85%), and then stop moving and curl their bodies [20]. In this non-moving state, metabolism is stopped. The results presented here reveal that *Bursaphelenchus*
*xylophilus* third-stage dispersal juveniles can undergo cryptobiosis. This state can be considered to exist between survival and death, with dehydration and rehydration before and after cryptobiosis crucial to survival. These findings are consistent with related reports that *B. xylophilus* may experience dehydration, cryptobiosis, and rehydration in winter. *Bursaphelenchus xylophilus* transform into third-stage dispersal juveniles in winter in northeast China’s mid-temperature zone, dehydrate as water content and host temperature decrease, and gradually enter cryptobiosis. In spring, individuals can rehydrate and recover with an increase in host water content and temperature.

Slow dehydration during cryptobiosis can promote nematode survival and prevent damage to in vivo protein structures [22]. Here, an initial 8% KCl solution was allowed to undergo evaporation, resulting in a gradual increase in KCl concentration. This caused the nematodes to dehydrate and enter cryptobiosis. If water evaporated too quickly or if *B. xylophilus* were directly added to KCl solution at high concentration, the nematodes exhibit decreased survival. We found 100% of *B. xylophilus* exhibit in vivo bubbles during rehydration when being transferred to distilled water from 40% KCl solution and dyeing test showed that these bubbles are fat granules (data not shown). The survival rate of *B. xylophilus* with bubbles decreased significantly. More complete understanding of this process is required for the development of effective control strategies, and future work should define the factors that can influence the survival rate of *B. xylophilus* third-stage dispersal juveniles during dehydration and rehydration stages.

Previous studies indicated that *B. xylophilus* can adapt to different environments in response to low temperature, drought, and high osmotic pressure, by adjusting osmosis on the body surface and decreasing dehydration by curling to decrease surface area [17]. Ions and micromolecular substances can freely exchange with each other through the surface cuticle due to the osmotic equilibrium mechanism as described in *C. elegans* [23,24], but micromolecular substances cannot pass through the cuticle [25]. An altered osmotic pressure environment can allow dehydration of *B. xylophilus*, further causing body curling. When transferred to a lower osmotic pressure environment, *B. xylophilus* third-stage dispersal juveniles in cryptobiosis can gradually recover. *Bursaphelenchus xylophilus* must be able to control exchange between in vivo and in vitro substances by osmotic regulation during dehydration and rehydration to reach osmotic equilibrium, indicating that osmotic regulation is crucial to cryptobiosis.

With a decrease in ambient temperature, pine trees must decrease their in vivo free water content to prevent freezing damage to cells. However, this may cause a rise in osmotic pressure of the ambient environment. Based on the change rules of in vivo water content and ions of pine trees under low temperatures, the results in this work indicate that the survival rate of *B. xylophilus* third-stage dispersal juveniles under low temperatures can be improved significantly by increasing the concentration of K^+^ ions in the ambient water environment and reaching cryptobiosis by osmotic regulation. Thus, the concentration changes of in vivo substances in pine trees under low temperature may help *B. xylophilus* resist low-temperature stress.

*Bursaphelenchus**xylophilus* was previously reported to tolerate temperatures below −5 °C [14]. Understanding the temperature resistance of this pest can be used to predict areas where it could spread. By 2017, pine wilt disease had spread to mid-temperature zones in China with an annual average temperature below 10 °C [17]. In this study, we found that *B. xylophilus* third-stage dispersal juveniles can survive under −80 °C, with a greater than 50% survival rate at −20 °C [26], lower than the previously described tolerated minimum temperature. Based on this, additional cold temperature and mid-temperature zones in China may be at risk for *B. xylophilus*.

The survival rate at −20 ℃ was greatly improved when *B. xylophilus* could enter cryptobiosis by osmotic regulation, reaching 92.1% [7]. This study demonstrates that *B. xylophilus* can enter cryptobiosis, *B. xylophilus* third-stage dispersal juveniles can resist low-temperature stress by cryptobiosis, and osmotic regulation is crucial to cryptobiosis [27]. These findings provide new theoretical support to predict the potential expanded distribution areas for *B. xylophilus* in the mid-temperature and cold temperature zones of China [28].

## Figures and Tables

**Figure 1 biology-10-00785-f001:**
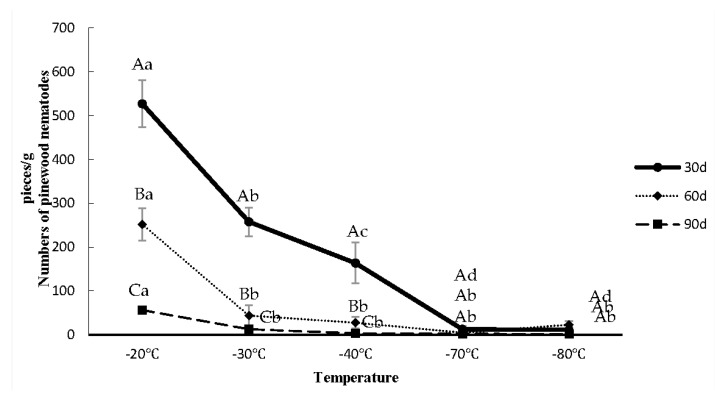
Survival of *Bursaphelenchus*
*xylophilus* third-stage dispersal juveniles in *P**inus tabuliformis* after being subjected to different temperature stress. Note: The first capital letter indicates a significant difference in *B. xylophilus* numbers between treatment times. The second lowercase letter indicates a significant difference in *B. xylophilus* numbers at different temperatures.

**Figure 2 biology-10-00785-f002:**
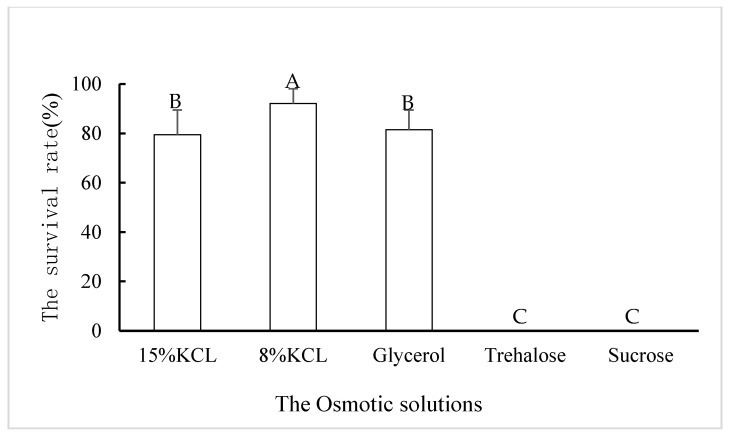
The survival rates of *Bursaphelenchus*
*xylophilus* third-stage dispersal juveniles after cryptobiosis by dehydration under −20 °C. Note: Different letters indicate significant differences in survival for different osmotic solutions.

**Figure 3 biology-10-00785-f003:**
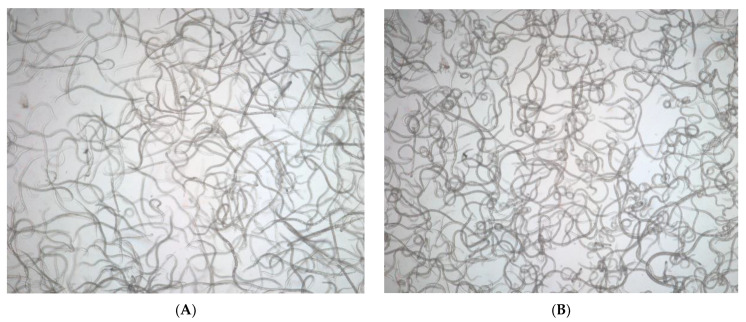
Dehydration and rehydration of *Bursaphelenchus*
*xylophilus* third-stage dispersal juveniles. (**A**) Rehydration; (**B**) start of dehydration; (**C**) early dehydration; (**D**) later dehydration; (**E**) complete dehydration; (**F**) magnified view showing the body surface of a dehydrated *B. xylophilus*.

## Data Availability

Not applicable.

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
