# Peer review of "Third-Stage Dispersal Juveniles of Bursaphelenchus xylophilus Can Resist Low-Temperature Stress by Entering Cryptobiosis"

_biology, 2021, doi:10.3390/biology10080785_

Round 1

Reviewer 1 Report

I suggest to improve the introduction section, again! The introduction do not provide sufficient background and do not include all relevant references.

Author Response

Thank you for your suggestion.We have improved our introduction section to introduce the research content and signification better.

Reviewer 2 Report

Dear authors, 

although you made some modifications improving your work, there are still some errors to be corrected. 
Additionaly, in my opinion you could add significant information about the nematodes. For instance, after the III dispersal juvenile stage (pre-dauer juveniles) entered a cryptobiosis state, how the cycle continues when spring conditions arrive?
Please take into account the comments along the text and modify it accordingly.

Good work,

REV

Author Response

Thank you for your suggestions.We have added the content of the cycle in spring in the introduction.

Reviewer 3 Report

Authors have improved the manuscript accordingly.

Author Response

Thank you for your suggestions.

Round 2

Reviewer 2 Report

Dear authors, 

although you improved your manuscript, in my opinion there are still some major corrections to be done. In particular, I advise you to use the appropriate nematological terms (such as juveniles J and not L as for larvae, a term no longer used for the juvenile stages of nematodes). The same for "extraction" of nematodes from wood: this is the appropriate word and not "isolation" or "removal".

Please, pay also particular attention to the References list because it has several errors to correct.

I hope you can consider my suggestions (see comments along the text) and improve your manuscript enough for publication.

I wish you good work and good health!

REV

Author Response

Dear reviewer

We are so grateful to your advice to our manuscript. All of your advice we have seen and used to improve our manuscript.

Line 15 we have replaced as your advice.

line62-71 we have replaced the letter L to J and DL3 DL4 to JIII  JIV.

But this paragraph we need to describe each stages of the B. xylophilus more clearly so we want to retain these content and improve.

line92 we have improve this sentence for correct 

line103 we have deleted the redundant point.

line 146-147 and 149 we have improve unit as reviews’ advice

And about the reference, we have communicated with the editor and according to the comments of reviewers and editors according to the format of the modification.

Thank you for your advice 

Round 3

Reviewer 2 Report

Dear authors, 

I'm so sorry but there are still some details requiring your attention. 
I hope you can consider my suggestions in order to improve your manuscript.

Best regards, 

REV

Author Response

Dear Reviewer:

    Thank you very much for your advice, which also reflects your serious and responsible attitude. We have revised the manuscript again according to your requirements and suggestions.

    Now there's only one point that needs to be made in line 75-79 that the Jâ…¢ nematodes form under adverse environmental conditions,which can be learned in the reference 17.And the beetles exist mainly to attract the Jâ…¢ nematodes to gather and induce them to chance to the Jâ…£ nematodes.

     Thank you again for your suggestions to help us correct. This time, we have modified according to your suggestions, and we hope it can be accepted.

   Best wishes.

Round 4

Reviewer 2 Report

Dear authors, 

Thank you for accepting my suggestions. The manuscript is now more suited for publication in Biology_MDPI.

Can you please make the final modifications, namely:

  • Page 3/9 Line 42 remove the repeated dot
  • P6 L188 replace maginified by magnified
  • P8 L273 add space before Forest
  • P8 L279 add space before June
  • P9 L300 add space before B. xylophilus
  • P9 L310 add space before (Coleoptera)

Best wishes, 
REV

This manuscript is a resubmission of an earlier submission. The following is a list of the peer review reports and author responses from that submission.

Round 1

Reviewer 1 Report

This paper investigates the cryptobiosis of Bursaphelenchus xylophilus. Please find below my suggestions:

Line 36. Bursaphelenchus as first mention must be written in full

Line 38. Pinus as first mention must be written in full

Line 54 and more. In vitro and in vivo must be written in latin

Line 64. Replace 3-age larva with third stage juveniles

Line 66. Add missing reference

Line 74. Larix capitol letter

Line 76. How many grams were used?

Line 77. Identification technique of nematodes must be included as well as tools used.

Line 85. the weight of the 5 pieces must be indicated.

Line 99. How many replications were done for each repetition?

Lines 88 to 109. More information on products used for the solutions must be provided.

Line 107. Information on the microscope used (brand, magnification, etc.) must be provided

Line 115. How does proportions were obtained?

Line 117. L. olgensis

Line 125. The density of B. xylophilus in p. tabuliformis after treatment should be displayed as figure or table and not included in the text.

Introduction and discussion sections must be more supported by literature investigation (references). The importance of Bursaphelenchus xylophilus should be highlighted. To improve the introduction section, I suggest to read this paper: d'Errico, G., Carletti, B., Schröder, T., Mota, M., Vieira, P., & Roversi, P. F. (2015). An update on the occurrence of nematodes belonging to the genus Bursaphelenchus in the Mediterranean area. Forestry: An International Journal of Forest Research88(5), 509-520.

Conclusions should be improved.

Reviewer 2 Report

The manuscript needs a complete revision of the English language, giving attention to the structure and length of sentences.

Scientific literature  reports  the definition “third-stage dispersal juveniles of Bursaphelenchus xylophilus” instead of “diffused 3-age Bursaphelenchus xylophilus”. Change in title and in the manuscript, maybe using an acronym (J3) after the first citation. Similarly, you could use the term “specimens” or “individuals” instead of pieces when you refer to B. xylophylus juveniles.

Figures 4 and 5 could be eliminated, as showing what already reported in Figure 3.

Figure 1 and 2 does not report any indicators of statistical diffences among the values, you should include them.

Other more specific observations are reported by following:

Line 18: in the manuscript there is no discussion of the potential utility of displayed results for the control of B. xylophilus.

Lines 66-69: in the Introduction  you should indicate the objectives of the study and not the results.

Line 73: do you mean “three trees” or “trees”? or “three what”?

Line 93: how the 300 dispersal juveniles were transferred to onto the concave glasses? Were they handly picked or pipetted in a water suspension? You should specify.

Lines 101-106: as the methodology was similar to that of experiment described at 2.3, microscopical observations could be includes in 2.3, as a part of the experiment here described.

Line 117: Larix olgensis, use italic and initial capital letter.

Line 109: in MM there is no specification of statistical analysis of data. Was it done? If yes, describe it and add statistical indicators in figures.

Lines 123-124: “those logs…”. This was previously described in MM, so remove from this position.

Line 130 and 145: Figures 1 and 2 do not report indicators of statistical differences among data: include.

Line 168 and 189: Figures 4 and 5 are not necessary, as nematode curling is already showed in Figures 3B and 3C, whereas fat granules are also visible in Figures 3E and 3F.

Lines 190-192: explain how findings of  this study “can provide a framework for prevention and control”.

Lines 229-232: the Conclusions section is too short. You could remove and make a Discussion and Conlusions section or, alternativel, move to the Conclusions some general statements reported in Discussion.

Reviewer 3 Report

Pine wilt disease, caused by the pinewood nematode (PWN, Bursaphelenchus xylophilus), is one the most devastating threats to conifer forests worldwide, and in particular in China. This study focused on the behaviour of the PWN under extreme conditions of low temperatures. The authors concluded that the nematodes undergo a cryptobiosis process allowing them to survive and recover the normal lifecycle under suitable conditions.

Dear authors, this work is relevant and deserves to be spread but it requires a huge amount of modifications. So, in my opinion the manuscript cannot be accepted for publication as it is in Biology_MDPI. The experimental design is incomplete or at least is not well detailed. While going through the experiment, it seems the work was done with P. tabuliformis alone. It is difficult to correlate the number of treatments [(4 tree species) x 3 trees x (3 areas of the trunk)]=36 samples with the results obtained; the age of trees is not mentioned ; the statistics used are not detailed and so on. The figures caption must be corrected and completed. Some references need attention and they are not well cited.  

Furthermore, some terms are confusing and usually not used. For example, will Diffused 3-Age Bursaphelenchus xylophilus correspond to the third dispersal juveniles (JIII)? That is, the correct terms “dispersal and propagative juveniles” were not used. The same is true for other nematological terms, which are not correct. From the point of view of the Nematology, this manuscript needs some improvements before being submitted for publication. As such, I added some comments along the text for your consideration.

I trust you can invest in improving this study and wish you good work.

Best regards,

REV

Reviewer 4 Report

The manuscript explores an interesting angle regarding the ecology and physiology of the pinewood nematode, thus providing interesting observations and results. However, the manuscript is very confusing to follow because of the poor English writing, therefore it should be reviewed by a native English-speaking person, before being re-considered for publication. There are too many errors and poor phrasing, however some examples follow, together with corrections and questions: Line 12: a common error: The article “the” is often used where it is unnecessary, thus “Pine wilt disease…”; please review the entire manuscript for this sort of mistake; Line 13: another common mistake: in the same paragraph using the past and present tense (“was”); please be coherent, throughout; L. 15: what do the authors mean by “diffuse”? And does “3-age B. x.” mean “3-year-old B.x.”? Change throughout; L. 37: “”Recently, B. xylophilus has continuosly…”; L.39-40: “… was greatly affected by PWD”; L. 48: why “multiple”? ºerhaps, many or numerous?; L. 54: “in vivo” and “in vitro” are Latin terms, should be in italic; L.63: what do the authors mean by “generative”? L.102: “dripped”? or dropped? L. 112: what do you mean by “separation”? L. 117: Larix olgensis: how does this tree appear here? L. 119: nematodes have juvenile (and not larval) stages; L. 123: again the present-past tense issue: the experiments should be described in the past tense; L. 154-155: what do you mean by “water excavation”? L. 158: “gathered” is not the best term; L. 165-170: this portion is very poorly written and confusing; L. 182: another misuse: “Tha paper selects..”; L. 211: what do the authors mean by “change rules of in vivo…”? Very confusing; L. 221: “In 2017, PWD,..”, not Bx disease;